# Acceptability and implementation potential of colorectal cancer screening and health literacy training: A qualitative study among general practitioners in deprived areas

Géraldine Cazorla[1], Niamh M. Redmond[2,3], Aurore Lamouroux[4,5], Alix Boirot[1], Michel Rotily[6], Maria Claudia Addamiano[2], Christian Balamou[7], Zineb Doukhi[8], Myriam Kaou[9], Françoise Couranjou[10], Cyrille Delpierre[2], Julien Mancini[1,11]*, Marie-Anne Durand[12,13]

1 Aix Marseille Univ, INSERM, IRD, ISSPAM, SESSTIM, Sciences Economiques & Sociales, Marseille, France, 2 EQUITY Research Team (Certified by the French League Against Cancer), Centre for Epidemiology and Population Health (CERPOP), INSERM, University of Toulouse, Toulouse, France, 3 ARC West/Bristol Medical School, University of Bristol, Bristol, United Kingdom, 4 Centre de Santé Universitaire – Espace Santé Aygalades, Assistance Publique - Hôpitaux de Marseille, Marseille, France, 5 Vaucluse Departmental Health Education Committee (CoDES 84), Avignon, France, 6 CEReSS-Health Service Research and Quality of Life Centre, Aix Marseille University, Marseille, France, 7 Centre Régional de Coordination du Dépistage des Cancers (CRCDC-AuRA), Auvergne-Rhônes-Alpes, Saint Étienne, France, 8 Unité d'Epidémiologie Clinique / CIC-EC 1426 – INSERM, Unité de Recherche Clinique de Robert Debré, Hôpital Universitaire Robert-Debré, Paris, France, 9 INSERM, UMR 1290 RESHAPE University Lyon 1, Lyon, France, 10 Assistance Publique - Hôpitaux de Marseille, Marseille, France, 11 Aix Marseille Univ, Assistance Publique - Hôpitaux de Marseille, Hôpital Timone, BioSTIC, Biostatistique et Technologies de l'Information et de la Communication, Marseille, France, 12 The Dartmouth Institute for Health Policy & Clinical Practice, Dartmouth College, Lebanon, New Hampshire, United States of America, 13 Unisanté, University Centre for Primary Care and Public Health, Department of Ambulatory Care, University of Lausanne, Lausanne, Switzerland

* julien.mancini@univ-amu.fr

⊙ OPEN ACCESS

**Data Availability Statement:** The raw data for this study consist of interview transcripts containing potentially identifying information. As participants

## Abstract

### Background

Colorectal cancer (CRC) is a significant contributor to cancer-related burden, ranking second in cancer mortality in France. Despite the proven survival benefits of systematic CRC screening, uptake remains suboptimal, particularly among people with limited health literacy (HL) and lower socioeconomic position. This study aimed to assess the acceptability of an e-learning training programme on HL and CRC screening among general practitioners (GPs) in deprived areas while also exploring strategies for its promotion and scale-up.

### Methods

A qualitative study nested within the DECODE cluster-randomised controlled trial (NCT04631692) across four French regions was conducted. Semi-structured interviews (phone or online) were carried out to capture opinions, experiences, and recommendations of GPs in the intervention arm. Thematic analysis, employing manual and NVivo coding, was performed.

did not consent to the sharing of their data beyond the study team, these data cannot be made publicly available. However, relevant de-identified excerpts from the transcripts are included in the article. In accordance with the trial's ethical approval and participant consent procedures, data will be available upon request and is subject to approval by the Trial Steering Committee, a supplementary approval by the University of Toulouse III Ethics Committee, and a data sharing agreement. For further information or data requests, please contact either the corresponding author or the trial manager, Dr. Niamh M. Redmond (niamh-maria. redmond@univ-tlse3.fr).

**Funding:** This research was financially supported by a grant (2020-006) from the French National Institute of Cancer (INCa) (https://en.e-cancer.fr) awarded to MAD and AL. No additional external funding was received for this study. The funder had no role in study design, data collection and analysis, decision to publish, or preparation of the manuscript.

**Competing interests:** The authors have read the journal's policy and one author of this manuscript have the following competing interests: MA-D has contributed to the development of Option Grid patient decision aids outside of the submitted work. EBSCO Information Services sells subscription access to Option Grid patient decision aids. She receives consulting income from EBSCO Health, and royalties outside of the submitted work. There are no patents, products in development or marketed products associated with this research to declare. This does not alter our adherence to PLOS ONE policies on sharing data and materials".

## Results

The majority of 22 GPs (16/22) found the training acceptable, informative, tailored to their knowledge needs and offering great flexibility of use. The module on HL garnered more interest than the one on CRC screening, as it addressed a relatively new area for many GPs. It facilitated reflection on patient communication techniques and identified areas for improvement in physician-patient interaction. A perceived gap in the training was the insufficient interactivity in both didactic and virtual group sessions.

## Conclusion

The findings of this study show high acceptability of the e-learning training by participants, indicating a potential for implementation, if kept concise, self-paced, asynchronous, and with a stronger focus on HL. The training helped GPs reflect on their practices, enhance HL knowledge, and improve patient communication strategies, leading some to adopt new techniques in daily interactions with low HL patients, beyond just screening.

## Introduction

Colorectal cancer (CRC) is a significant contributor to the global cancer burden; ranking third in terms of incidence and second in mortality worldwide. In France, it is the third most commonly diagnosed cancer and the second most common cause of cancer-related deaths after lung cancer, with over 17,000 fatalities annually [1, 2]. Undergoing a Faecal Immunochemical Test (FIT) and colonoscopy (when the FIT test is positive) reduces CRC-related mortality [3].

The French national CRC-screening programme was launched in 2009 and targets people aged 50 to 74. Eligible persons receive standardized invitation letters, every 2 years, to visit their general practitioner (GP), specialist or pharmacist to obtain a home and self-administered FIT test [4]. However, CRC screening test uptake remained suboptimal at 34.3% in 2021–22 and far below European targets of between 45% (minimum) and 65% (desirable) [5, 6]. Screening uptake in France is socially graded [7, 8], decreases with increasing levels of deprivation and is lower among those in socially deprived areas [7, 9]. Similar to global trends, lower screening uptake is associated with lower socioeconomic position with significant geographic disparities [10–12]. There is some evidence that patient navigators can facilitate participation in CRC screening in low socio-economic areas [13] but these could be costly in terms of both time and financial costs.

As deprivation levels rise, screening participation decreases, exacerbating social inequities in health [11, 14]. For example, in France and across Europe, higher educational attainment is associated with lower mortality rates for nearly all cancer types [15]. This educational gradient in health may be partly related to health literacy as defined as "the degree to which individuals have the capacity to obtain, process, and understand basic health information and services needed to make appropriate health decisions" [16]. HL is an important determinant of health and is socially distributed [13]. Low or limited HL is associated with poorer general health status, poorer ability to interpret health messages, higher disease burden and higher mortality [17–20] underscoring its importance as a key factor in influencing CRC screening uptake [21].

The role of inadequate HL is well documented and increasingly recognized as a barrier that significantly impacts CRC screening knowledge, beliefs, and behaviours [22, 23]. Individuals with lower HL are less likely to seek and comprehend information about CRC screening and

have lower self-efficacy for screening [24]. Addressing HL is therefore essential to improving outcomes and minimising health disparities [18]. In cancer prevention, improving HL can reduce social disparities in screening and potential disparities in survival [25].

GPs play a crucial role in addressing HL as they often are the first point of contact within the healthcare system [26, 27]. However, they are often unaware of patients' literacy/HL challenges or may not know how to address them [28]. Most studies on HL have focused on patient education, with less emphasis on the communication skills and practices of healthcare practitioners [26, 28]. While GP communication is recognized as a factor in CRC screening [29, 30], GPs frequently encounter difficulties in conveying evidence-based information about CRC screening to patients with limited HL [31].

Despite some evidence that HL interventions can improve CRC screening in certain populations [31, 32] research remains limited and their impact on health inequities is not thoroughly examined [33]. Further research into the development and implementation of evidence-based health literacy interventions to improve cancer outcomes are recommended [34]. To this end, we developed and conducted an e-learning training programme on HL and CRC screening for GPs practising in deprived areas across four regions of France namely Auvergne-Rhône-Alpes, Île-de-France, Occitanie, and Provence-Alpes-Côte d'Azur. The areas targeted by the project in these four regions have CRC screening rates below the national average and were selected based on a European Deprivation Index (EDI) of 4 or 5, indicating the most deprived areas. The training program was part of a multicentre, cluster-randomised controlled trial [35] which aimed to increase CRC screening uptake and also included a patient-facing intervention (pictorial brochure and video) conducted during consultations.

In this trial, we included several qualitative studies to gain a deeper understanding of the integration of the intervention among the targeted individuals and the participating GPs. Here, we present here a nested qualitative study with GPs who were part of the intervention group.

The objective of this study was to evaluate the acceptability of an e-learning training programme on HL and CRC screening for GPs practising in deprived areas. We aimed to assess whether this training could be implemented among a broader population of GPs and explore strategies to promote its widespread implementation across France and within the healthcare system.

## Methods

### Study design

We conducted a cross-sectional qualitative study with GPs randomised to the intervention arm to assess the acceptability and implementation potential of the HL and CRC training among a broader population of GPs.

We reported our results using the Consolidated Criteria for Reporting Qualitative Research (COREQ) (S1 Appendix).

We used semi-structured interviews to enable a more detailed analysis and a richer insight into the experience of the e-learning training among GPs in the intervention arm. This qualitative approach allows for a more in-depth and flexible exploration of stakeholders' perspectives compared to a survey or other quantitative methods. It also enables researchers to probe deeper into specific topics or questions [36].

We reported our results using the Consolidated Criteria for Reporting Qualitative Research (COREQ) (S1 Appendix). We also adhered to the 'Eight Big-Tent Criteria for Excellent Qualitative Research' [37], a flexible pedagogical model that distinguishes between means (methods and practices) and outcomes (ends). It introduces eight key markers of quality: (a) worthy

topic, (b) rich rigour, (c) sincerity, (d) credibility, (e) resonance, (f) significant contribution, (g) ethics, and (h) meaningful coherence.

## Recruitment of general practitioners

In the DECODE trial, GPs were selected in each region on the basis of their practice address, which had to correspond to an EDI of 4 or 5. GP names were randomly selected from the list of registered GPs working in these areas. Initial contact with GPs was made via email or telephone with details about the trial. In cases where this initial approach was unsuccessful, we used a convenience, snowballing method, making use of our research network contacts. Confirmation of participation was obtained by email and additional contact details were collected.

GPs who agreed to participate were block-randomised, stratified by region and cluster (GP practice or GP), to a combined intervention (HL and CRC training + brochure and video for eligible patients), or usual care. GPs in the same practice were allocated to the same trial group to avoid contamination. If a participating GP left their practice or withdrew from the trial, a replacement GP was recruited and assigned to the same group.

## Intervention

The GP training, adapted from a previously tested intervention [31], consisted of a 2-hour e-learning session followed by a 1-hour booster session. The first hour of the training session was completed independently by participating GPs in the intervention arm. It was video-based with 4 modules on i) the DECODE trial, ii) CRC screening, ii) HL information, and; iv) strategies for effective communication with patients. Secondly, a c-MOOC (Connectivist–Massive Online Learning Course) format was organised, with interactive discussions between GPs who had watched the 4 video modules and a trained facilitator. This promoted engagement through small group discussions, role-playing and teach-back. A 1-hour c-MOOC "booster" training session was held six months later with feedback on recruitment and aggregated screening uptake rates. See the published protocol for more details [35]. Each GP in the intervention group received reimbursements for completing the initial 2-hour training (€150) and the booster session (€75). Inputs from professional stakeholders, including members of the Community Advisory Board, the Trial Steering Group, and a separate GP focus group were incorporated to appraise and improve the training content [38].

## Setting and participants of the qualitative study

All GPs assigned to the intervention group were invited to participate in the qualitative interviews. We contacted GPs using a purposive sampling method to ensure representation of both male and female practitioners, across each region and including those that had withdrawn or retired from the main trial.

In May 2023, 3 weeks after the participant recruitment period had finished, GPs in the intervention arm (n = 32) were contacted by email to participate in a semi-structured interview. Based on previous experience of qualitative studies of this type, we expected roughly 20 participants to reach thematic saturation thematic saturation [39]. To ensure a balance across regional centres we completed 22 GP interviews (4–5 participants per region). All participants provided fully informed written consent.

## Data collection

The interview guide (S2 Appendix) comprised 21 questions and gathered information on GPs' views regarding the acceptability and implementation potential of the HL and CRC training

among a broader population of GPs. The topics covered included (1) motivations to participate in the training programme; (2) perception of the training programme (format, content, room for improvement); (3) impact on knowledge and clinical practice and (4) attitudes towards implementation. The guide was developed based on Normalization Process Theory (NPT) [40, 41] a theoretical model that assists in understanding the integration of complex interventions into standard healthcare practices. NPT has been used to develop and evaluate the implementation of various healthcare interventions in primary care settings, including training initiatives [42, 43]. It revolves around four core theoretical constructs (Table 1).

Interviews were performed by telephone or online (via Zoom™ or Teams™ platforms) by two researchers (GC and AB). All interviews were audio-recorded, transcribed verbatim and checked for accuracy. Data were pseudonymized during the transcription process of the audio files. Transcripts were not returned to participants for approval.

## Data analysis

A framework analysis with inductive methods [44, 45] was initially conducted to identify any emerging codes. NPT was then applied as an analytical lens to examine the data, as it provides a suitable framework for exploring participants' views on how interventions impact their work and can be integrated into routine practice. This framework has been used in studies on training initiatives, shared decision-making, and conversation aids [43, 46–48]. This theory aligned well with our objective of assessing the acceptability of the training while identifying recommendations for the sustainability of future training and practices. The coding process was guided by the core constructs and components of NPT. Codes pertaining to the implementation of the patient-directed intervention were not included, as they were the focus of a separate study and article. Analysis involved manual and NVivo coding performed separately by the two researchers whose finding were subsequently merged. Any consensus or discrepancies on key issues were discussed and new and important themes identified. Different perspectives were also explored. Key analytical themes were discussed within the research team to ensure external validity.

For each quotation, a code was assigned to indicate the GP practice region, for example, GP-OCCI refers to a GP interviewed in the Occitanie region.

## Ethics statement

Ethical approval for the study 2021–349 was obtained by the Research Ethics Committee (CER) at the University of Toulouse III Paul Sabatier on March 8, 2021 and DECODE trial

**Table 1. Normalization process theory (NPT) constructs in association with the implementation of the HL and CRC e-learning training for GPs.**

| | |
|---|---|
| Sense-making or coherence | Sense-making or coherence is about understanding the utility and significance of a complex intervention through individual and communal interpretation<br>*What is the purpose of receiving training on HL and CRC*? |
| Cognitive participation | Participation involves the cognitive involvement that either encourages or hinders users' acceptance and commitment to the intervention<br>*Which factors influence GP's engagement in the e-learning training*? |
| Action | Action involves collective efforts that either facilitate or impede the intended use of the intervention by everyone involved<br>*How do GPs work to optimise the impact of the e-learning training for implementing the patient-facing intervention*? |
| Monitoring | Monitoring involves communal and individual assessments of the intervention's impact.<br>*How do GPs appraise the e-learning training in terms of improving knowledge and practices and what do they recommend for scaling it up*? |

was registered on the 17th of November 2020 on ClinicalTrials.gov with the trial registration number 2020-A01687-32. The consent form signed by GPs for the trial also served as the basis for their involvement in the qualitative study. In essence, GPs who consented to participate in the trial also consented to participate in the qualitative study.

## Results

The Interviews were conducted between June and September 2023 and lasted between 30 and 60 minutes. GC interviewed GPs from the Ile de France (IDF) and Occitanie (OCCI) regions, while AB interviewed GPs from the Provence Alpe-Côte-d'Azur (PACA) and Auvergne Rhône Alpes (AURA) regions.

### General practitioners (GPs) characteristics

Of the 32 GPs contacted, 10 did not respond, either because they had already withdrawn from the study, retired or moved to another region. Table 2 describes the characteristics of the 22 GPs interviewed. The GPs interviewed consisted of 13 women and 9 men, aged between 31 and 64 years, with an average age of 39 years. They were selected from four regions across France, with 5 or 6 participants from each region. These GPs mainly worked in multidisciplinary health centres situated in socioeconomically disadvantaged areas.

### Major themes according to NPT constructs

The main themes, organised according to the four NPT constructs, are summarised in Table 3. All themes were derived during the analysis from codes generated from interview transcript.

**Table 2. Characteristics of the general practitioners (GPs) interviewed (n = 22).**

| Variable | Number of GPs | |
|---|---|---|
| | N | % |
| *Location (regions)*: | | |
| *AURA* | 6 | 27% |
| *IdF* | 6 | 27% |
| *Occitanie* | 5 | 23% |
| *PACA* | 5 | 23% |
| *Type of working environment*: | | |
| *Individual practice* | 0 | |
| *Group practice* | 4 | 18% |
| *Multi-professional health centre* | 12 | 55% |
| *Health centre* | 6 | 27% |
| *Gender–male* | 9 | 41% |
| *No of years since qualified as a GP at the time of recruitment*: | | |
| *0–5* | 8 | 36.5% |
| *5–10* | 8 | 36.5% |
| *+10* | 6 | 27% |
| *Participation in sessions of the DECODE intervention training*: | | |
| *Initial video training* | 22 | 100% |
| *Interactive session* | 22 | 100% |
| *Booster session* | 15 | 68% |
| *Type of interview*: | | |
| *Telephone* | 11 | 50% |
| *Video conference* | 11 | 50% |

**Table 3. Summary of major themes according to NPT constructs.**

| NPT construct | Major themes |
|---|---|
| Sense-making | GPs felt the training would be helpful for better communication with patients with low HL. It also helped with CRC screening, a common but challenging topic in preventive care. |
| Cognitive participation | GPs found the training format engaging. Interest in HL was another key driver of participation. |
| Action | GPs found virtual group sessions essential but recommended they be more grounded in real-life situations. |
| Monitoring | GPs recommended expanding the training to more GPs, provided it stays brief, more interactive, and focused on HL. |

### Major theme related to sense-making—"What is the purpose of receiving training on HL and CRC?"

For most intervention GPS, the training was considered well-aligned with their needs. Their decision to participate in the study, including the trial's training programme, was primarily driven by its resonance with the situations faced by physicians practising in deprived areas, as a significant proportion of their patients have low HL and/or difficulties with French.

*"Well, it's. . . it's really connected to our. . . to our reality especially in neighbourhoods like this, where we work."*

GP-AURA

In this context, raising awareness about the concept of HL was seen as crucial for improving communication with low HL patients.

*"(. . .) we're not adequately sensitised to it [HL] and to the various techniques that can be used to effectively convey the message, including to patients with low levels of literacy."*

GP-OCCI

Improving CRC screening, a common topic during consultations yet challenging to integrate into preventive practices, was another important factor in participating in the trial and its training programme. While some GPs prioritised it, others acknowledged that, despite its importance, they did not do enough.

*"We know we're not doing well on the colorectal cancer screening test and that, well, we need to look for solutions to improve, especially within our populations."*

GP-IDF

The opportunity to receive training, resources, and support to engage patients with CRC screening was perceived as a real motivator given the public health importance of the issue.

*"Participation in colorectal cancer screening is within our public health objectives [. . .] if I could find tools that allow me to further improve this participation, because once again, if we can do prevention, it's still better."*

GP-OCCI

Although the relevance of the training was widely agreed upon, some respondents suggested that it might only appeal to already convinced GPs. They questioned whether other physicians, particularly those with patients exhibiting higher HL levels, would show interest in the topic. A comment likely related to the fact that most participants work as staff in healthcare facilities, with some participating in other projects focused on reducing health inequalities.

For most, the training did not disrupt their schedules. However, a few respondents (n = 3) mentioned that a lack of time due to high workload could be an issue. The "work environment under increasing strain", often mentioned during interviews, and the numerous training offers they receive, may contribute to this observation.

The issue of time prompted one respondent to indirectly mention the issue of reimbursement, knowing that each GP could receive up to €225 for completing the training.

*"Time is already hard to find (laughs), I would say, on a daily basis, and (. . .) in fact, in a way, time is money."*

GP-IDF

The issue of money, although never presented as the primary reason for receiving the training, surfaced as a need for recognition for the extra time required despite their busy schedules.

## Major theme related to cognitive participation—"Which factors influence GPs' engagement in the e-learning training?"

Overall, GPs had a rather vague memory of the online training–a situation likely more related to the time elapsed since participation than to a lack of interest. Indeed, up to 25 months (minimum: 10 months) had passed since the training as the interviews could not be conducted during the recruitment period. However, after a small effort of recollection, most respondents (n = 16) spontaneously described factors that facilitated their engagement in the training, such as finding it acceptable, informative, tailored to their knowledge needs and very flexible to use.

*"(. . .) it really set the framework on what was expected of us, on, well, also explaining what the purpose of the study was to our patients. So, you know, it's. . . everything was very clear."*

GP-AURA

A set of factors was cited as favouring acceptance and participation in the training programme both in terms of form and content. The training format was deemed 'accessible', didactic', 'clear', 'concise', and well-suited to GPs' time limitations. For example, GPs were asked to view the 1-hour didactic course (4 modules) within a week, so that the second hour could be organised, however they could go back and review the videos if they wanted to. Only one GP was not convinced by the training's structure, noting that they found it somewhat restrictive to participate multiple times due to its division into several modules.

In terms of training content, the modules focusing on HL and strategies to improve physician-patient communication emerged as the primary factors driving GPs' engagement, cited either for deepening existing knowledge or introducing new concepts. More than a third of GPs (n = 8) and certain colleagues (e.g., nurses, medical assistants) acknowledged they were unfamiliar with the concept of HL.

*"I see my colleagues, my friends who are also doctors, for whom the term health literacy (. . .) is not known, and. . . they surely have patients who have this low level of literacy."*

GP-IDF

The modules on HL were seen as an opportunity to reflect on one's own practices and those of colleagues. In particular, some respondents realised that many GPs could face comprehension issues directly linked to low HL without being fully aware of it.

*"I had a colleague who did it [the training], and (. . .) he also couldn't imagine that (. . .) patients could leave our consultations without understanding anything."*

GP-OCCI

For some GPs, participating in the training provided a sort of theoretical validation for what they had already been doing intuitively. Although not necessarily familiar with the term HL, they were already using certain techniques and tools to facilitate better understanding of information such as "rephrasing with simpler words", "speaking in more adapted language", or "using images".

*"It allowed me to have a perhaps slightly more theoretical insight into things that I feel like, you know, practising, let's say (laughs)."*

GP-IDF

Regardless of their familiarity with the concept, GPs considered HL the most useful aspect of the training programme with advice standing out to them, for instance *"that you should have a maximum of three messages in a discussion, how to rephrase with simple words, well, things like that.*" (GP-IDF). However, three GPs expressed regret that the HL module was not comprehensive enough. Among them, one did not find the training very satisfactory, noting that it seemed more like a briefing than a training and they did not feel adequately trained in HL. This raised the question of finding the right balance between too much and not enough.

The module on CRC screening did not garner the same level of acceptance as the one on HL due to the abundance of information available on the topic. Nearly half of the GPs (n = 10) found the existing resources on CRC via various channels to be adequate and sufficient. Some respondents however, considered it useful to update their epidemiological knowledge on CRC.

*"There were some stats at the beginning to recall the context of colorectal cancer. [. . .] it allowed me to refresh my memory on this screening, even in my mind."*

GP-IDF

### Major theme related to action—"How do GPs work to optimise the impact of the e-learning training for implementing the patient-facing intervention?"

The online interactive sessions, intended to facilitate GPs training experience by bridging theory with practice, received mixed reviews. While some believed they provided a platform for collaboratively testing what they have learned and sharing experiences, others found them

lacking in interactivity. Positive aspects highlighted included the opportunity to "have a connection with a human at one point", discuss challenges, "share tips among peers", "get motivated to restart inclusions", and receive knowledge reminders.

The 1-hour c-MOOC, organised shortly after the initial didactic session to empower GPs to effectively discuss CRC screening with their patients, received little feedback. When feedback was provided, the practical scenarios were viewed as too few and too simplistic. For instance, the role playing was described as "too basic" and lacking realism compared to the challenges encountered daily with a patient population typically having low HL.

The 15 GPs who participated in the 1-hour c-MOOC booster training session held 6 months later often had a vague recollection of it. Those who remembered it saw it as an opportunity to discuss the patient-facing intervention (video and brochure) they had implemented in the interim to enhance its effectiveness.

*"I found it really good that we could give feedback on what was working or not working, and be able to readjust. And also, the fact that it was with other practitioners participating in the study, um. . . it was very good because there was this exchange of peers, (. . .) for example, I remember someone saying how they presented the study. So it helped to be able to present it yourself to your patients, well, we shared a bit of the difficulties, and then the solutions."*

GP-IDF

One third of respondents (n = 5) highlighted that the refresher training was a motivating factor in giving "courage and energy to include patients".

*"At first, I was quite motivated to recruit patients, I found that patients were quite happy to be offered to participate in the study (. . .) But I don't know, after that, I lost track if [. . .] I think we had another video call in the autumn and it kind of rebooted me and I found it not bad for that (laughs)."*

GP-PACA

A few GPs also saw the six-month refresher training as an opportunity to update their knowledge.

*"I had continued to learn things, especially about these issues of, the keywords, the fact of reminding that the test was simple, practical, free. [. . .] it allows to complete nevertheless because, whether we realise it or not, in the first [training], we didn't necessarily remember everything."*

GP-OCCI

Finally, the personalised nature of the training, which provided GPs with their recruited patients and screening rates was appreciated.

*"So, it's not just like: we send you an email or a video or. . . but having someone there, I thought that was great. And then giving us. . . well, our own figures, the results of the patients we had. . . who were available, that kind of thing".*

GP-IDF

Providing feedback to GPs also fostered a competitive spirit and healthy competition among participants.

### Major theme related to monitoring—"How do GPs appraise the e-learning training in terms of improving knowledge and practices and what do they recommend for scaling it up?"

The training programme, including the peer exchanges, emphasised the opportunity for GPs to become more familiar with the concept of HL and reflect on their communication with patients. It enabled some GPs to realise the disparity between their communication and patient understanding.

*"I had realised that sometimes, I gave a lot of information and maybe I overwhelmed people, so it had helped me a lot and made me question my way of explaining things to people. . . it's not easy to change, but, you know, to try to say to yourself: 'Well, here are three main ideas and too bad for the others, three main ones, because otherwise in the end, nothing is retained."*

GP-OCCI

Realising this communication gap prompted more than a quarter of respondents (n = 6) to adopt new communication techniques such as feedback that help enhance interactions with patients both during the study and beyond, and not only in the context of CRC screening.

*"On everything related to feedback, asking the patient to summarise what they understood. [. . .] Now, I do it, and as a result, I do it not only on this issue. I do it, for example, on understanding prescriptions, which I didn't do before. [. . .] It really allowed me to improve my daily practice and not just on the question of screening, ultimately on my way of communicating with the patient."*

GP-AURA

The few reservations expressed about the training mainly related to its lack of interactivity and remote format, underscoring the importance of having virtual group exchanges to solidify the knowledge gained from the didactic modules. This sentiment was echoed by some, albeit with some nuance, highlighting the challenges of organising meetings between GPs, even virtually, due to time limitations, increased workloads and health measures in place due to the COVID-19 pandemic.

*"In these MOOC or e-learning type trainings, there's this weird aspect of being alone in front of your screen. (. . .) But at the same time (. . .) I think for healthcare professionals, what's easier is being able to choose your own time, your own schedule (. . .) when you can do it. In fact (. . .) for the practical session, we were supposed to be five or six, something like that, and in the end, we were only three because nobody could free up their schedule."*

GP-IDF

Interactivity was perceived differently by the respondents. For some, the issue with the didactic modules was not so much about the online format itself, but with their inherent lack of interactivity which could have been addressed by incorporating more engaging learning activities such as quizzes and mind maps. For others, it was primarily the limited use of real-world situations that posed a problem in fully internalising the teachings of the training and applying them in practice.

*" I remember there were, you know, the terms given, some tips. But I couldn't put them into practice. I didn't feel well-equipped enough because we didn't work on them (. . .) Well, we had some exchanges, but it's not enough, actually. We needed to have workshops with. . . with clinical scenarios or, you know, with role-playing."*

GP-PACA

Despite these few drawbacks, all GPs found the training useful and tailored to their needs, especially the section dedicated to HL, praised by nearly two-thirds (n = 14). As a result, most would strongly recommend the training to another general practitioner, though opinions varied on who should be prioritised for the training. More than half (n = 12) felt the training should target all GPs, regardless of their practice settings or patient populations, while others (n = 5) believed it should focus specifically on GPs whose patients have low HL levels.

*" I believe you have only seen* [as part of the study] *doctors who work in slightly more disadvantaged areas, and therefore we are all a little more concerned about this which may not be the case for all of our colleagues. But despite everything, inevitably, one day or another, we are confronted with it [low HL levels], and probably more often than we think."*

MG-AURA

All interviewed GPs recommended expanding the training on a larger scale to enhance communication with patients and contribute more actively to CRC screening. Suggestions for fostering GPs' buy-in and commitment while maximising the impact of the training programme are summarised in Table 4.

**Table 4. Suggested improvements to scale up the e-learning training programme.**

| Training content-related issues | Suggested training content improvement |
|---|---|
| CRC module deemed less useful (ample existing resources) | Emphasise HL over technical aspects of CRC screening |
| Hands-on scenarios considered too few and simplistic | Include more role-playing, complexify scenarios, and base them on real-life practice situations |
| **Training format-related issues** | **Suggested training format improvement** |
| Overall lack of interactivity of the training | Enhance interactivity across the training |
| • First hour didactic session (4 modules) | Incorporate MCQs, mind maps, and quizzes to assess knowledge retention, enhance focus, and facilitate learning |
| • Second hour c-MOOC (learner-driven dialogue) | Offer hybrid training sessions whenever possible (e.g., face-to-face sessions) to facilitate interactions |
| **Other identified issues** | **Suggested improvements** |
| • Time and money-related issues | • Keep the training brief and self-paced to allow flexibility<br>• Offer incentives, even symbolic, to acknowledge the commitment and time invested<br>• Value participation in studies and their impact on practice (e.g., increased patient screenings) in the ROSP [payment for public health objectives] |
| • Potential lack of interest from some GPs in HL | Promote HL training through catchy communication, for instance:<br>• *"Do your patients truly understand you*?" MG-OCCI or<br>• *"(. . .) a patient with low health literacy (. . .) we can't just spot it by looking at them, that it's often invisible, and we can't know it."* MG-PACA |

The following channels were considered for disseminating the training: 1) social security, primary health insurance funds; 2) initial or continuous medical training (for example, Continuous Professional Development [CPD], Family Medicine Society, Professional Training Funds) notably through webinars; 3) the Board of Physicians (via its newsletters); 4) departments of general medicine (via mailing lists); 5) regional health agencies; 6) medical doctors' unions; 7) medical institutions' social media platforms.

## Discussion

### Main findings

The results suggested a high level of acceptability of the training programme, with GPs finding it informative, tailored to their knowledge needs and offering great flexibility of use. This suggests strong potential for widespread implementation as evidenced by the nearly unanimous recommendation to endorse this training for other general practitioners.

The majority of GPs expressed a notable preference for the HL module, underscoring the importance and relevance of increasing awareness about a concept that was unfamiliar or new to many of them. This was particularly pressing as their patient population primarily comprised individuals with limited HL and/or difficulties with French. It helped reflect on how to communicate with patients and identify areas for improvement in physician-patient communication. Our findings suggest, that integrating new communication methods (e.g., simplifying messaging, checking for understanding, asking patients to repeat information, etc.) may have resulted in a change in practice beyond the promotion of CRC screening. Although the modules on HL-based communication strategies garnered more interest than the one focusing on CRC screening, some GPs appreciated the opportunity to refresh their epidemiological knowledge.

Regarding the format, participants stressed the importance of maintaining a self-paced and asynchronous approach to accommodate time constraints. The perceived gap in the training was a lack of interactivity in both didactic and group sessions. The fully remote format of the training also posed some issues. However, implementing training in an in-person or hybrid format was seen as complicated due to cost and physician availability concerns as well as an increased workload due to the COVID-19 period.

### Findings in relation to the results of other studies

Our findings support the literature from the US and UK [26, 49] highlighting GPs' lack of confidence in addressing issues around HL and emphasise the importance of building communication skills [50]. GPs' communication style and interpersonal competence are more critical to patients than technical competence during the CRC diagnostic process [51, 52], potentially influencing early versus late-stage diagnosis. Placing an even greater emphasis on HL and communication strategies was deemed a top priority by participating GPs as observed in another study conducted in the French context [53], which underscored the importance of promoting patient-physician dialogue through clear and motivating explanation of the screening procedure. However, while training GPs in cancer risk communication can improve their counselling behaviours, it does not always translate into higher screening rates among patients with limited HL [54].

Receiving feedback about their screening rates was viewed as encouraging promotion of screening, consistent with findings from several reviews that demonstrated increased uptake as a result of this intervention [55–57]. Provider feedback interventions have also proven effective in increasing breast and cervical screening uptake [56, 58].

Suggestions to enhance interactivity in didactic modules included integrating engaging learning activities (e.g., quizzes, mind maps). This aligns with research findings indicating that a balanced combination of information and interactivity, along with communication examples and serious games, influences GPs' adoption of web-based learning and improves learning outcomes [24, 59, 60]. A study on MOOC instructors' strategies to encourage self-directed learning (SDL) found that they motivated learners by identifying the value and needs of learning, boosting self-efficacy, and offering incentives. To sustain motivation during tasks, they used engaging teaching methods, well-designed materials, timely feedback, and social interaction [61].

Regarding online group sessions, GPs recommended making them more practice-oriented and utilizing interactive methods such as "teach-back", feedback, role-plays, and small group discussions which have shown effectiveness in other studies for improving patient comprehension [29, 49, 62–64] and increasing CRC screening uptake [50, 55]. These insights and recommendations will help update and enhance the training to maximize its impact.

Some GPs would have preferred the group sessions to be conducted face-to-face rather than online to increase interaction. However, organising a sufficient number of GPs for group sessions was already challenging, despite its importance for sharing experiences and implementing practice changes. Similar difficulties have been previously observed where only four out of the twelve eligible general practices approached were recruited for training, resulting in a low recruitment rate and a reduction in training time from one day to 3 hours to accommodate the busy schedules of GPs. This is coherent with our findings which indicated a clear preference for keeping the training concise, but contrary to a systematic review demonstrating that face to face programmes teaching communication skills to GPs are effective if they last for at least one day [63].

The majority of respondents (n = 17) did not consider remuneration as a motivating factor or an incentive for their participation in the intervention, as evidenced in other studies [56], specifying that salaried GPs in health centres cannot be individually remunerated. However, some participants emphasised that even a small payment was important to recognize their commitment and time, aligning with studies in France suggesting the need for incentives to enhance GP involvement [50, 65]. However, there is insufficient evidence to assess the effectiveness of provider incentives in increasing screening rates for breast, cervical, or colorectal cancers [56].

As with other study recommendations from the US and France [26, 29], all participants recommended disseminating the training to the entire GP population, both through initial medical education and continuing education. However, further research is needed to determine if whether all GPs or those with patient populations exhibiting low levels of HL are more effectively targeted to increase CRC screening participation rates. The results of the DECODE trial will go some way to answering this.

## Strengths and limitations

Study limitations include selection and social desirability biases, as many participants already showed an interest in HL despite varying degrees of familiarity with the concept. This may have influenced their participation in the training programme. In addition, the majority of participants are healthcare facility staff, with some involved in other projects aimed at reducing health inequalities, also potentially leading to an overrepresentation of GPs keen to promote HL.

Although this is a qualitative study and the findings cannot necessarily be generalised, the high participation rate in the interviews (69%), the rich qualitative data collected and the

consistency of responses provides confidence that our findings can lead to recommendations in terms of the content and design of an e-learning programme on HL and CRC screening that could be acceptable and relevant for GPs with a view to scaling up.

## Conclusion

The findings of this study show a high level of acceptance of the e-learning training by participants, suggesting a potential for implementation, particularly if a stronger focus on HL is added. More generally GPs' training seems to be better received if it focusses on little-known but important concepts such as HL, rather than on a medical practice such as screening. Our main results indicate that the training enabled GPs to reflect on their practices and improve their knowledge of HL and strategies for better patient communication. This led some GPs to adopt communication techniques in their daily interactions with low HL patients, extending beyond the subject of screening.

To sustain the lasting impact of the training, we recommend annual online refresher sessions to reinforce learning and keep skills up-to-date. These sessions should include peer discussion groups for sharing experiences and addressing challenges through case studies. Additionally, brief, asynchronous micro-learning modules should be developed, allowing GPs to review key concepts at their convenience. Finally, newsletters and distance-learning need assessment surveys should be used to tailor content to GPs' evolving needs, supporting ongoing self-directed learning.

## Supporting information

**S1 Appendix. COREQ checklist.**
(PDF)

**S2 Appendix. GP interview guide.**
(DOCX)

## Acknowledgments

We would like to extend our warmest thanks to all the participants of the DECODE trial, the GPs, and our colleagues at the four Regional Cancer Screening Coordination Centres, as well as Fecop (Federation of Primary Care Coordinated Multi-professional Practices) and our other partners who assisted with this trial. Without their support, this study would not have been possible. We also wish to express our gratitude to our colleagues A. Martinez, R. Grami, and N. Allami, who stepped in for the Clinical Research Associates during their absences. Additionally, we are grateful to all the members of our Community Advisory Board for their expertise and insights, and to our Steering Committee members for their invaluable input and advice. This study, nested within the DECODE trial, was funded by the French National Institute of Cancer.

## Author Contributions

**Conceptualization:** Niamh M. Redmond, Aurore Lamouroux, Julien Mancini, Marie-Anne Durand.

**Data curation:** Niamh M. Redmond, Aurore Lamouroux, Marie-Anne Durand.

**Formal analysis:** Géraldine Cazorla, Niamh M. Redmond, Alix Boirot.

**Funding acquisition:** Aurore Lamouroux, Michel Rotily, Christian Balamou, Marie-Anne Durand.

**Investigation:** Géraldine Cazorla, Niamh M. Redmond, Aurore Lamouroux, Alix Boirot, Marie-Anne Durand.

**Methodology:** Aurore Lamouroux, Marie-Anne Durand.

**Project administration:** Niamh M. Redmond, Aurore Lamouroux, Maria Claudia Addamiano, Christian Balamou, Zineb Doukhi, Myriam Kaou, Julien Mancini, Marie-Anne Durand.

**Resources:** Niamh M. Redmond, Aurore Lamouroux, Julien Mancini, Marie-Anne Durand.

**Software:** Niamh M. Redmond.

**Supervision:** Niamh M. Redmond, Aurore Lamouroux, Michel Rotily, Christian Balamou, Cyrille Delpierre, Julien Mancini, Marie-Anne Durand.

**Validation:** Aurore Lamouroux, Marie-Anne Durand.

**Visualization:** Niamh M. Redmond, Alix Boirot.

**Writing – original draft:** Géraldine Cazorla, Alix Boirot.

**Writing – review & editing:** Géraldine Cazorla, Niamh M. Redmond, Aurore Lamouroux, Alix Boirot, Michel Rotily, Maria Claudia Addamiano, Christian Balamou, Zineb Doukhi, Myriam Kaou, Françoise Couranjou, Cyrille Delpierre, Julien Mancini, Marie-Anne Durand.

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
