## [Decision Letter · Decision Letter 0]

28 Oct 2024

PONE-D-24-32539Acceptability and implementation potential of colorectal cancer screening and health literacy training: A qualitative study among general practitioners in deprived areasPLOS ONE

Dear Dr. Cazorla, Thank you for submitting your manuscript to PLOS ONE. After careful consideration, we feel that it has merit but does not fully meet PLOS ONE’s publication criteria as it currently stands. Therefore, we invite you to submit a revised version of the manuscript that addresses the points raised during the review process.

We look forward to receiving your revised manuscript.

Kind regards,

Alvaro Galli

Academic Editor

PLOS ONE

Journal Requirements:

1. When submitting your revision, we need you to address these additional requirements. Please ensure that your manuscript meets PLOS ONE's style requirements, including those for file naming. The PLOS ONE style templates can be found at https://journals.plos.org/plosone/s/file?id=wjVg/PLOSOne_formatting_sample_main_body.pdf and https://journals.plos.org/plosone/s/file?id=ba62/PLOSOne_formatting_sample_title_authors_affiliations.pdf 2. We note that the grant information you provided in the ‘Funding Information’ and ‘Financial Disclosure’ sections do not match.  When you resubmit, please ensure that you provide the correct grant numbers for the awards you received for your study in the ‘Funding Information’ section. 3. Thank you for stating the following in the Competing Interests section: "I have read the journal's policy and one author of this manuscript have the following competing interests: MA-D has contributed to the development of Option Grid patient decision aids. EBSCO Information Cervices sells subscription access to Option Grid patient decision aids. She receives consulting income from EBSCO Health, and royalties. No other competing interests declared." Please confirm that this does not alter your adherence to all PLOS ONE policies on sharing data and materials, by including the following statement: ""This does not alter our adherence to  PLOS ONE policies on sharing data and materials.” (as detailed online in our guide for authors http://journals.plos.org/plosone/s/competing-interests).  If there are restrictions on sharing of data and/or materials, please state these. Please note that we cannot proceed with consideration of your article until this information has been declared.  Please include your updated Competing Interests statement in your cover letter; we will change the online submission form on your behalf. 4. We note that you have indicated that there are restrictions to data sharing for this study. For studies involving human research participant data or other sensitive data, we encourage authors to share de-identified or anonymized data. However, when data cannot be publicly shared for ethical reasons, we allow authors to make their data sets available upon request. For information on unacceptable data access restrictions, please see http://journals.plos.org/plosone/s/data-availability#loc-unacceptable-data-access-restrictions.  Before we proceed with your manuscript, please address the following prompts: a) If there are ethical or legal restrictions on sharing a de-identified data set, please explain them in detail (e.g., data contain potentially identifying or sensitive patient information, data are owned by a third-party organization, etc.) and who has imposed them (e.g., a Research Ethics Committee or Institutional Review Board, etc.). Please also provide contact information for a data access committee, ethics committee, or other institutional body to which data requests may be sent. b) If there are no restrictions, please upload the minimal anonymized data set necessary to replicate your study findings to a stable, public repository and provide us with the relevant URLs, DOIs, or accession numbers. Please see http://www.bmj.com/content/340/bmj.c181.long for guidelines on how to de-identify and prepare clinical data for publication. For a list of recommended repositories, please see https://journals.plos.org/plosone/s/recommended-repositories. You also have the option of uploading the data as Supporting Information files, but we would recommend depositing data directly to a data repository if possible. Please update your Data Availability statement in the submission form accordingly. 5. In the online submission form, you indicated that [Requests for further information can be directed to the corresponding author.] All PLOS journals now require all data underlying the findings described in their manuscript to be freely available to other researchers, either 1. In a public repository, 2. Within the manuscript itself, or 3. Uploaded as supplementary information.This policy applies to all data except where public deposition would breach compliance with the protocol approved by your research ethics board. If your data cannot be made publicly available for ethical or legal reasons (e.g., public availability would compromise patient privacy), please explain your reasons on resubmission and your exemption request will be escalated for approval. 6. Please include your full ethics statement in the ‘Methods’ section of your manuscript file. In your statement, please include the full name of the IRB or ethics committee who approved or waived your study, as well as whether or not you obtained informed written or verbal consent. If consent was waived for your study, please include this information in your statement as well. 7. Please include captions for your Supporting Information files at the end of your manuscript, and update any in-text citations to match accordingly. Please see our Supporting Information guidelines for more information: http://journals.plos.org/plosone/s/supporting-information.

Reviewers' comments:

Reviewer's Responses to Questions

**Comments to the Author**

1. Is the manuscript technically sound, and do the data support the conclusions?

Reviewer #1: Partly

Reviewer #2: Yes

2. Has the statistical analysis been performed appropriately and rigorously? 

Reviewer #1: N/A

Reviewer #2: Yes

3. Have the authors made all data underlying the findings in their manuscript fully available?

Reviewer #1: Yes

Reviewer #2: Yes

4. Is the manuscript presented in an intelligible fashion and written in standard English?

Reviewer #1: Yes

Reviewer #2: Yes

5. Review Comments to the Author

Reviewer #1: Thank you for the opportunity to review this manuscript, which addresses an important field for research, namely how health literacy is vital for participation in CRC screening. However, the manuscript has several elements that it addresses. While the introduction concerns CRC screening, the study that has been conducted is about how GPs evaluate a learning intervention about health literacy and CRC screening. It is a bit confusing that the introduction is more about the umbrella study, rather than the issue at stake in this particular qualitative study.

A second overall comment is about the choice of qualitative interviews and how the data from the interviews is presented in the results section. While the idea of doing a qualitative interview study is good, it is not clear from the results what this choice of method has contributed to, which could not have been identified through using a survey with open ended questions. I would urge the authors to clarify why this design was chosen and what its contribution is. Perhaps this manuscript would benefit from being presented as an evaluation study rather than a qualitative study?

Some more detailed comments:

104-107: The introduction is very well written. However, instead of talking about the number of qualitative studies nested within your RCT, could you rather explain why you do the study presented here? What was the current study expected to contribute to the overall study?

115-116: “We reported our results using the Consolidated Criteria for Reporting Qualitative Research (COREQ) (S1. Appendix COREQ checklist)”

COREQ is a list of issues that needs to be addressed in each article. However, it is not sufficient to create a high-quality article. I would suggest that the authors of this article look at more qualitative articles to see how they bring depth to their findings. Also, I would suggest reading Sarah Tracy (2010) article on “Big 8 criteria for qualitative research”.

Something looks strange in the document here:

147 Setting and participants of the qualitative study

148 Recruitment of GPs to the qualitative study

150: what is the purpose of including gender of GP as a factor for the sample? Do you have literature suggesting that gender would be important for the theme of the study?

154: How did you evaluate that 20 participants were necessary for saturation? What kind of saturation are you referring to here?

158: what is meant by “implementation potential”? Is this the potential for implementing the GP training among GPs? Or is it about implementation of HL knowledge among patients?

Data analysis: was the analysis inspired by any analytical approach? Which literature did you use when deciding on inductive thematic analysis? Could you explain more in detail how you used NPT during your analysis?

181: I would suggest deleting this sentence, it is unnecessary to explain this.

187: Characteristics of participants should be moved to the methods section.

Table 3. Summary of major themes according to NPT constructs is difficult to grasp due to the explaining sentences being very long, complex and using many concepts.

210: t factor?

221: you write that “A few respondents (n=3)”. What is the purpose of counting how many said what in a qualitative study? This is also done in line 229 (and several places in the results section) where it says that “The majority of GPs (17 out of 22)…”. This makes me question if the results are analyzed through a qualitative approach, or if this is rather an evaluation of the HL training. The results are solely descriptive, even when using the theoretical framework of NPT. This is a weakness of the study, since the analysis does not utilize the qualitative aspect of the data.

Reviewer #2: Positive Aspects:

1. The study tackles the low CRC screening rates, especially in deprived areas, and emphasizes the role of health literacy, which is crucial for improving public health outcomes.

2. The qualitative study offers deep insights into GP perspectives, providing valuable input for future interventions.

3. Focusing on GP training is strategic, as they are key influencers in promoting CRC screening.

4. The paper offers practical suggestions for improving training programs, which could aid in scaling up such initiatives.

Suggestions for Improvement:

1. The 68% attendance in booster sessions and participants' fading memories suggest the need for repeated reinforcement training. Could the authors discuss the necessity of such sessions and how future training could be enhanced to ensure lasting effects?

2. Further analysis on the relationship between health literacy and CRC screening uptake could be beneficial. Are there specific health literacy aspects that significantly influence screening behavior, and how can they be targeted in training?

3. More context on the current CRC screening situation in the study area, including any identified initiatives or barriers, would provide a clearer backdrop for the training program.

Overall, the manuscript provides important insights into GP training's role in promoting CRC screening and health literacy. With some additional discussion on the points raised, the study could significantly contribute to the field.

6. PLOS authors have the option to publish the peer review history of their article (what does this mean?). If published, this will include your full peer review and any attached files.

Reviewer #1: No

Reviewer #2: No

---

## [Author Response · Author response to Decision Letter 0]

11 Dec 2024

Responses to academic editor and reviewers

We thank the editor and reviewers for their valuable feedback

Academic editor’s comments: 

Answer: Regarding the affiliations, the SESSTIM has its own publication charter (starting with AIX-Marseille Univ). Please let us know how to proceed. For more information, feel free to contact the trial manager Dr Niamh Redmond niamh-maria.redmond@univ-tlse3.fr

Answer: Upon resubmission, we aimed to include all the required information in both the Financial Disclosure and under “Funding Information» sections but we experienced difficulty identifying where exactly these two sections are located. Here is the correct information in response to your comment: 

“This research was supported by a grant (2020-006) from the French National Institute of Cancer (INCa) https://en.e-cancer.fr/ awarded to MAD and AL. The funders had no role in study design, data collection and analysis, decision to publish, or preparation of the manuscript.

Please let us know if there are any further issues with this. 

3. Thank you for stating the following in the Competing Interests section: "I have read the journal's policy and one author of this manuscript have the following competing interests: MA-D has contributed to the development of Option Grid patient decision aids. EBSCO Information Cervices sells subscription access to Option Grid patient decision aids. She receives consulting income from EBSCO Health, and royalties. No other competing interests declared."

Answer: The following statement: “This does not alter our adherence to PLOS ONE policies on sharing data and materials” was included both in the cover letter and at the end of manuscript. 

4. We note that you have indicated that there are restrictions to data sharing for this study. For studies involving human research participant data or other sensitive data, we encourage authors to share de-identified or anonymized data. However, when data cannot be publicly shared for ethical reasons, we allow authors to make their data sets available upon request. 

For information on unacceptable data access restrictions, please see http://journals.plos.org/plosone/s/data-availability#loc-unacceptable-data-access-restrictions. 

Answer: We have updated our Data Availability Statement as follows: 

The corresponding author has full access to the original dataset supporting this article. In accordance with the trial’s ethical approval and participant consent procedures, data will only be available ‘on request’ and is subject to approval by the Trial Steering Committee, a supplementary approval by the University of Toulouse III Ethics Committee and a data sharing agreement. Please contact either the corresponding author or the trial manager Dr Niamh M Redmond (niamh-maria.redmond@univ-tlse3.fr) for further information. 

5. In the online submission form, you indicated that [Requests for further information can be directed to the corresponding author.]

Answer: Our Data Availability Statement clarifies the restrictions we are required to comply with regard to data sharing: The corresponding author has full access to the original dataset supporting this article. In accordance with the Trial’s Ethical approval and participant consent procedures, data will only be available ‘on request’ and is subject to approval by the Trial Steering Committee, a supplementary approval by the University of Toulouse III Ethics Committee and a data sharing agreement. Please contact either the corresponding author or the trial manager Dr Niamh M Redmond (niamh-maria.redmond@univ-tlse3.fr) for further information. 

Answer: The full ethics statement is now included in the ‘Method’ section as follows: 

“Ethical approval for the study 2021-349 was received from the local research ethics committee, the CER (comité d'éthique de la recherche) at the University of Toulouse III Paul Sabatier, France (ref 2021–349, dated 8th March 2021, category MR004) and the DECODE trial was registered on the 17th of November 2020 on ClinicalTrials.gov with the trial registration number 2020-A01687-32. Informed written consent for participation in the trial and the qualitative studies was obtained from all participating general practitioners.”

Answer: A section titled “Supporting information” was included at the end of the manuscript, listing supporting information captions. The supporting information files at the end of the manuscript has now been reviewed and updated to match accordingly.

Reviewers' comments:

Reviewer #1: 

1. Thank you for the opportunity to review this manuscript, which addresses an important field for research, namely how health literacy is vital for participation in CRC screening. However, the manuscript has several elements that it addresses. While the introduction concerns CRC screening, the study that has been conducted is about how GPs evaluate a learning intervention about health literacy and CRC screening. It is a bit confusing that the introduction is more about the umbrella study, rather than the issue at stake in this particular qualitative study.

Answer: Thank you for the feedback. To clarify the research question and the objective of the study, we have replaced lines 102–110 (from 'To add to the body of knowledge > widespread implementation' in the introduction section) with the following sentence:

 “To this end, we developed and conducted an e-learning training program on HL and CRC screening for GPs practising in deprived areas across four regions of France. This training program was part of a multicentre, cluster-randomised controlled trial (REF 30) which aimed to increase CRC screening uptake and also included a patient-facing intervention (pictorial brochure and video) conducted during consultations. In this trial, we included several qualitative studies to gain a deeper understanding of the integration of the intervention among both targeted patients and participating GPs.

Here, we present a nested qualitative study with GPs in the intervention group. The objective of this study was to evaluate the acceptability of an e-learning training programme on HL and CRC screening for GPs practising in deprived areas. We aimed to assess whether this training could be implemented among a broader population of GPs and explore strategies to promote its widespread implementation across France and within the healthcare system”.

2. A second overall comment is about the choice of qualitative interviews and how the data from the interviews is presented in the results section. While the idea of doing a qualitative interview study is good, it is not clear from the results what this choice of method has contributed to, which could not have been identified through using a survey with open ended questions. I would urge the authors to clarify why this design was chosen and what its contribution is. Perhaps this manuscript would benefit from being presented as an evaluation study rather than a qualitative study?

Answer: We chose a qualitative study because it is a recognized and validated method for assessing the acceptability and implementation potential of an intervention (https://pmc.ncbi.nlm.nih.gov/articles/PMC3758883/). Qualitative studies nested with a randomised controlled trial (RCT) are common in public health research and add complementary value to the often purely quantitative aspect of the RCT. This qualitative approach was used to complement the quantitative data collected to assess the acceptability of the training.

In the design section, we included the following in order to provide details on the choice of method and included the reference above: 

“We used semi-structured interviews to enable a more detailed analysis and a richer insight into the experience of the e-learning training among GPs in the intervention arm. This qualitative approach allows for a more in-depth and flexible exploration of stakeholders' perspectives compared to a survey or other quantitative methods. It also enables researchers to probe deeper into specific topics or questions. In particular, GPs could directly suggest ways to improve the training in terms of structure, format, content, and methods of dissemination to facilitate its future implementation”.

Some more detailed comments:

3. 104-107: The introduction is very well written. However, instead of talking about the number of qualitative studies nested within your RCT, could you rather explain why you do the study presented here? What was the current study expected to contribute to the overall study?

Answer: Thank you for your positive feedback on the introduction. We have refined the study’s objective and clarified its intended broader contribution as follows at the end of the introduction section: 

“The objective of this study was to evaluate the acceptability of an e-learning training programme on HL and CRC screening for GPs practising in deprived areas. We aimed to assess whether this training could be implemented among a broader population of GPs and explore strategies for promoting its widespread implementation across France and within the healthcare system”. 

4. 115-116: “We reported our results using the Consolidated Criteria for Reporting Qualitative Research (COREQ) (S1. Appendix COREQ checklist)”

COREQ is a list of issues that needs to be addressed in each article. However, it is not sufficient to create a high-quality article. I would suggest that the authors of this article look at more qualitative articles to see how they bring depth to their findings. Also, I would suggest reading Sarah Tracy (2010) article on “Big 8 criteria for qualitative research”.

Answer: We ensured that our study adhered to the highest standards of qualitative research by designing a robust qualitative study nested within the trial. Our team includes several researchers with over 15 years of experience conducting qualitative studies, ensuring methodological rigour and expertise.

The use of the COREQ checklist is widely used in qualitative research and is listed as a recommended checklist for research quality and transparency in qualitative studies on the Equator network ; https://www.equator-network.org/reporting-guidelines/coreq/. 

The COREQ checklist was first published in 2007 (https://doi.org/10.1093/intqhc/mzm042) and, in addition, PLOS recommends its use and inclusion for all submitted qualitative studies for review : https://journals.plos.org/plosone/s/submission-guidelines#loc-qualitative-research. 

Thank you for the recommended reference to the “Big 8 criteria for qualitative research”. After reviewing the eight criteria of this pedagogical model, we found that our study aligns well with the key markers of quality. Therefore, we are including this reference. 

"We also adhered to the 'Eight Big-Tent Criteria for Excellent Qualitative Research' (https://journals.sagepub.com/doi/10.1177/1077800410383121) a flexible pedagogical model that distinguishes between means (methods and practices) and outcomes (ends). It introduces eight key markers of quality: (a) worthy topic, (b) rich rigour, (c) sincerity, (d) credibility, (e) resonance, (f) significant contribution, (g) ethics, and (h) meaningful coherence."

5. Something looks strange in the document here:

147 Setting and participants of the qualitative study

148 Recruitment of GPs to the qualitative study

Answer: "Thank you for your observation. For clarity, we deleted the Level 2 heading 'Recruitment of GPs to the qualitative study' and kept only 'Setting and participants of the qualitative study.'

6. 150: what is the purpose of including gender of GP as a factor for the sample? Do you have literature suggesting that gender would be important for the theme of the study?

Answer: Although we identified several studies (https://pmc.ncbi.nlm.nih.gov/articles/PMC6731795/
https://pubmed.ncbi.nlm.nih.gov/8332153/
https://pubmed.ncbi.nlm.nih.gov/26765146/) showing that GP gender was independently associated with the quality of care regarding process indicators for cancer screening, we did not investigate this aspect due to the small sample size. We reported gender as well as age as basic socio-demographic information on the sample recruited. 

7. 154: How did you evalu

---

## [Decision Letter · Decision Letter 1]

7 Jan 2025

Acceptability and implementation potential of colorectal cancer screening and health literacy training: A qualitative study among general practitioners in deprived areas

PONE-D-24-32539R1

Dear Dr.Cazorla,

We’re pleased to inform you that your manuscript has been judged scientifically suitable for publication and will be formally accepted for publication once it meets all outstanding technical requirements.

Kind regards,

Alvaro Galli

Academic Editor

PLOS ONE

Additional Editor Comments (optional):

Reviewers' comments:

Reviewer's Responses to Questions

**Comments to the Author**

1. If the authors have adequately addressed your comments raised in a previous round of review and you feel that this manuscript is now acceptable for publication, you may indicate that here to bypass the “Comments to the Author” section, enter your conflict of interest statement in the “Confidential to Editor” section, and submit your "Accept" recommendation.

Reviewer #2: (No Response)

2. Is the manuscript technically sound, and do the data support the conclusions?

Reviewer #2: (No Response)

3. Has the statistical analysis been performed appropriately and rigorously? 

Reviewer #2: (No Response)

4. Have the authors made all data underlying the findings in their manuscript fully available?

Reviewer #2: (No Response)

5. Is the manuscript presented in an intelligible fashion and written in standard English?

Reviewer #2: (No Response)

6. Review Comments to the Author

Reviewer #2: (No Response)

7. PLOS authors have the option to publish the peer review history of their article (what does this mean?). If published, this will include your full peer review and any attached files.

Reviewer #2: No

---

## [Editor Report · Acceptance letter]

21 Jan 2025

PONE-D-24-32539R1 

PLOS ONE

Dear Dr. Cazorla, 

I'm pleased to inform you that your manuscript has been deemed suitable for publication in PLOS ONE. Congratulations! Your manuscript is now being handed over to our production team.

Kind regards, 

on behalf of

Dr. Alvaro Galli 

Academic Editor

PLOS ONE